# Non-Zoonotic Transmission of Sporotrichosis: A Translational Study of Forty-Three Cases in a Zoonotic Hyperendemic Area

**DOI:** 10.3390/jof10090610

**Published:** 2024-08-27

**Authors:** Juliana Nahal, Rowena Alves Coelho, Fernando Almeida-Silva, Andréa Reis Bernardes-Engemann, Anna Carolina Procópio-Azevedo, Vanessa Brito de Souza Rabello, Rayanne Gonçalves Loureiro, Dayvison Francis Saraiva Freitas, Antonio Carlos Francesconi do Valle, Priscila Marques de Macedo, Manoel Marques Evangelista Oliveira, Margarete Bernardo Tavares da Silva, Rosely Maria Zancopé-Oliveira, Rodrigo Almeida-Paes, Maria Clara Gutierrez-Galhardo, Maria Helena Galdino Figueiredo-Carvalho

**Affiliations:** 1Laboratório de Micologia, Instituto Nacional de Infectologia Evandro Chagas (INI), Fundação Oswaldo Cruz (Fiocruz), Rio de Janeiro 21040-900, Brazil; juliananahal04@gmail.com (J.N.); rowena.alves@ini.fiocruz.br (R.A.C.); fernando.almeida@ini.fiocruz.br (F.A.-S.); andreaengemann@yahoo.com.br (A.R.B.-E.); vanessa.brito@ini.fiocruz.br (V.B.d.S.R.); rayloureiro0701@gmail.com (R.G.L.); rosely.zancope@ini.fiocruz.br (R.M.Z.-O.); rodrigo.paes@ini.fiocruz.br (R.A.-P.); 2Laboratório de Pesquisa Clínica em Dermatologia Infecciosa, Instituto Nacional de Infectologia Evandro Chagas (INI), Fundação Oswaldo Cruz (Fiocruz), Rio de Janeiro 21040-900, Brazil; anna.azevedo@ini.fiocruz.br (A.C.P.-A.); dayvison.freitas@ini.fiocruz.br (D.F.S.F.); antonio.valle@ini.fiocruz.br (A.C.F.d.V.); priscila.marques@ini.fiocruz.br (P.M.d.M.); maria.clara@ini.fiocruz.br (M.C.G.-G.); 3Laboratório de Micologia, Instituto Oswaldo Cruz (IOC), Fundação Oswaldo Cruz (Fiocruz), Rio de Janeiro 21040-900, Brazil; manoel.marques@fiocruz.br; 4Serviço de Vigilância em Saúde, Instituto Nacional de Infectologia Evandro Chagas (INI), Fundação Oswaldo Cruz (Fiocruz), Rio de Janeiro 21040-900, Brazil; margarete.tavares@ini.fiocruz.br

**Keywords:** *Sporothrix brasiliensis*, *S. globosa*, *S. schenckii*, non-zoonotic transmission, non-wild-type strain, antifungal susceptibility

## Abstract

Over the past two decades, zoonotic sporotrichosis transmitted by naturally infected cats has become hyperendemic in Rio de Janeiro, Brazil. *Sporothrix brasiliensis* is the main agent involved. However, there are other forms of transmission of sporotrichosis. The aim of this study was to evaluate and associate the epidemiological, clinical and therapeutic data and the susceptibility of *Sporothrix* spp. to antifungal drugs in 43 non-zoonotic sporotrichosis cases. Forty-three clinical strains of *Sporothrix* were identified by partial sequencing of the calmodulin gene. An antifungal susceptibility test of amphotericin B, terbinafine, itraconazole, posaconazole and isavuconazole was performed according to the broth microdilution method. Most patients were male (55.8%). Regarding the source of infection, 21 patients (48.8%) reported trauma involving plants and/or contact with soil. *Sporothrix brasiliensis* was the predominant species (n = 39), followed by *S. globosa* (n = 3) and *S. schenckii* (n = 1). *Sporothrix brasiliensis* was associated with all the sources of infection, reinforcing previous data showing the presence of this species in environmental sources, as well as with all the clinical forms, including severe cases. One clinical strain of *Sporothrix brasiliensis* was classified as a non-wild-type strain for amphotericin B and another for itraconazole. *S. schenckii* was classified as non-WT for all the antifungals tested. In this context, it is important to emphasize that non-zoonotic sporotrichosis still occurs in the state of Rio de Janeiro, with *S. brasiliensis* as the main etiological agent, primarily associated with infections acquired after traumatic inoculation with plants and/or soil contact, followed by *S. globosa* and *S. schenckii*. In addition, non-WT strains were found, indicating the need to monitor the antifungal susceptibility profile of these species. It is crucial to investigate other natural sources of *S. brasiliensis* to better understand this fungal pathogen and its environment and host cycle.

## 1. Introduction

Sporotrichosis is a neglected implantation mycosis with worldwide distribution. It is the main subcutaneous mycosis in Latin America and is endemic in Brazil. The disease is caused by thermodimorphic fungi belonging to the genus *Sporothrix.* Their natural habitat is vegetation, decaying wood and soil. The classic form of sporotrichosis transmission is associated with traumatic inoculation by contaminated materials in natural environment, presenting risks to the main related occupational activities such as agriculture, floriculture, farming and logging that facilitate exposure to the fungus and its inoculation into the skin and/or subcutaneous tissue [1,2].

This mycosis is a public health problem in Brazil, especially in the state of Rio de Janeiro, and depending on the severity of the disease it can lead to hospitalizations and deaths [3]. It is currently considered a hyperendemic disease involving humans and animals [4,5]. Since 1998, zoonotic transmission of sporotrichosis in Rio de Janeiro state has been the main source of infection for humans, with *S. brasiliensis* being the main etiological agent of this mycosis [6,7,8]. It can occur through scratches, bites or even direct contact with secretions from infected cats [9]. However, the classical route of transmission has also been observed. The manipulation of plants or soil was a risk factor for the acquisition of human sporotrichosis in about 16.6% of the 1848 cases reported between 1997 and 2007 in Rio de Janeiro, Brazil [10]. In addition, other non-zoonotic forms of transmission may occur in endemic areas, such as post-tattoo sporotrichosis [11].

Studies involving patients who acquired the infection through non-zoonotic transmission are scarce in the scientific literature. For this reason, the current study aimed to evaluate the epidemiological, clinical and therapeutic aspects of non-zoonotic transmission, identify the species involved and determine the antifungal susceptibility profile of *Sporothrix* clinical strains isolated from human cases of non-zoonotic sporotrichosis from 2005 to 2020, providing a translational approach to patients’ clinical and laboratory data.

## 2. Materials and Methods

### 2.1. Ethical Approval

This study was approved by the Research Ethics Committee of Instituto Nacional de Infectologia Evandro Chagas da Fundação Oswaldo Cruz (INI/Fiocruz), the reference center for clinical assistance of sporotrichosis in Rio de Janeiro state, under the number CAAE: 53338521.6.0000.5262 on 28 December 2021.

### 2.2. Elaboration of the Database and Clinical Strains

A database was generated from the Central Electronic Medical Record System (SIPEC) based on human cases registered under the International Classification of Disease (ICD-10) of sporotrichosis between 1998 and 2020 and followed at INI/Fiocruz. The medical records were analyzed individually in order to detect cases of non-zoonotic sporotrichosis to ensure that the included patients had had no contact with animals, especially cats. The records were explored to obtain the variables of interest (sex, age, occupation, comorbidity, form of transmission, clinical form, lesion topography, treatment, change in treatment, adjuvant treatment and clinical outcome).

Based on the medical records review, clinical strains previously identified as *Sporothrix* spp. and stored in the Mycology Laboratory of INI/Fiocruz were recovered in potato dextrose agar medium (PDA; Difco, Becton-Dickinson and Company, Sparks, MD, USA) and incubated at 25 °C for 7 days to evaluate the viability and purity of the colonies. Subsequently, the clinical strains were subcultured in brain heart infusion agar medium (BHI; Difco, Becton-Dickinson and Company, Sparks, MD, USA) and incubated at 35 °C for 7 days, aiming to obtain thermal dimorphic conversion to the yeast-like phase.

### 2.3. Molecular Identification

DNA extraction was performed according to Ferrer et al. [12]. Clinical strains were identified by amplification and partial sequencing of the calmodulin gene. The primers Forward CL1 (5′-GA (GA) T (AT) CAA GGA GGC CTT CTC-3 and Reverse CL2A (5′-TTT TTG CAT CAT GAG TTG GAC-3′) were used [13]. Then, the PCR products were purified using a commercial purification kit (QIAquick^®^; Qiagen, Valencia, CA, USA). DNA sequencing was performed on the PDTIS/Fiocruz platform using the ABI-3730 sequencer (Applied Biosystems, Waltham, MA, USA). Sequences were edited using the Sequencher™ version 4.9, aligned and analyzed with the MEGA program version 7.0, and compared by BLAST with sequences deposited in the NCBI/GenBank database. The phylogenetic tree based on these data was inferred using the maximum likelihood method and significance was verified using the bootstrap confidence test with 1000 replications [14].

### 2.4. Antifungal Susceptibility Testing

All the strains were submitted to the broth microdilution method according to the reference document recommended by the Clinical and Laboratory Standards Institute [15]. Amphotericin B (AMB), itraconazole (ITR), posaconazole (POS), isavuconazole (ISA) and terbinafine (TRB) (Sigma-Aldrich, St. Louis, MO, USA) were tested. The inoculum was prepared from a 7-day-old PDA culture. Next, the cells were harvested in RPMI medium and diluted to 0.4–5 × 10^4^ cells/mL. Plates were incubated at 35 °C for 72 h and the minimal inhibitory concentrations (MIC) were determined according to CLSI M38-3Ed recommendations [15]. The tests were performed in duplicate.

The epidemiological cutoff values (ECVs) proposed for *S. brasiliensis* and *S. schenckii* were used [16]. MIC values for AMB, ITR, POS and TRB above 4 µg/mL, 2 µg/mL, 2 µg/mL and 0.12 µg/mL, respectively, were classified as non-wild-type (non-WT) strains. ISA does not have defined ECVs for *S. brasiliensis*, *S. schenckii* and *S. globosa*.

### 2.5. Statistical Analyses

The variables sex, age, form of infection, place of origin, occupation, clinical form, clinical specimens, fungal isolation data, treatment regimen and clinical evolution were stored in a Microsoft Excel^®^ database 2013. Relative and absolute frequencies were calculated. Descriptive statistical analyses were performed with the Statistical Package for the Social Sciences (SPSS), version 17.0, for Windows, to obtain the MIC range, MIC50 and MIC90 values and geometric mean. MIC50 and MIC90 values correspond to the minimal inhibitory concentration of the antifungal able to inhibit the growth of 50 and 90% of all fungal isolates, respectively.

## 3. Results

### 3.1. Analysis of Socio-Demographic, Epidemiological and Clinical Data of Patients

Forty-three cases were included, the first from 2005. Most patients were male (55.8%). Regarding age, six patients were under 18 years, 25 patients were aged between 19 and 59 years, and 12 patients were elderly (≥60 years). The mean age of the patients was 43 years (range: 11 to 77 years).

Among the 43 patients, 32 (74.4%) had some condition/comorbidity such as high blood pressure, diabetes mellitus, heart disease, chronic obstructive pulmonary disease (COPD), hyperthyroidism, hypothyroidism, dyslipidemia, alcoholism and smoking. Two patients (4.7%) were immunosuppressed (HIV/AIDS and kidney transplant). Eight patients presented more than one comorbidity.

The occupations/professions of the 43 patients were distributed as follows: retired (14%); housewives (14%); students (14%); service workers and trade sellers (9.3%); no information (7%); mid-level technician (7%); painting workers (7%); service workers in general (4.7%); teacher (4.7%); and masonry structure workers, bus drivers, auto mechanics, structural assembly workers in civil works, members of the armed forces, police and fire service, commerce operators in stores and markets, and economists and people outside the labor market, who represented 2.3% each.

Twenty-one patients (48.8%) reported episodes of trauma with plants and/or soil contact. Among the reports, the practice of gardening was observed, such as handling twigs, mango branches and rose thorns, and actions such as weeding and fertilizing land, in addition to falls/traumas in patients who practiced hiking and camping. Twelve patients (27.9%) were unaware of the method of inoculation and stated that they had not had any type of trauma involving plants and/or contact with animals. Four patients (9.2%) reported possible insect bites (mosquito and wasp) and two (4.7%) stated previous trauma while guaranteeing the absence of contact with animals. In addition, one patient (2.3%) presented the injury after being attacked by another person, suffering injuries to their the face, one (2.3%) mentioned having been bitten by a human on the neck during a fight, one (2.3%) was injured on an iron gate at his house and one (2.3%) described an incident with a fish bone.

The clinical forms included lymphocutaneous (65%), fixed cutaneous (23.3%), disseminated cutaneous (4.7%), disseminated (2.3%) and extracutaneous (4.7%) sporotrichosis.

Regarding treatment response and clinical outcomes, ITR was used in twenty-seven patients, of whom twenty-two (81.5%) achieved clinical cure, four (14.8%) were lost to follow-up and one (3.7%) was still undergoing treatment. TRB was the second most used drug, prescribed for three patients, all of whom progressed to clinical cure.

Overall, thirty-five patients (81.4%) achieved clinical cure, while six (13.9%) were lost to follow-up and two (4.7%) were still undergoing treatment.

The epidemiological, clinical, therapeutic and outcome data are summarized in Table 1.

Table 2 summarizes the distribution of lesion topography according to the sporotrichosis clinical forms, involving the lymphocutaneous and the fixed cutaneous forms; the upper limbs were the most affected sites.

Among the occupations/professions of the 43 patients included in this study, it was observed that students, retired people and housewives were the groups most affected (Table 3).

Itraconazole was the most prescribed drug, followed by TRB. Some patients used physical methods associated or not with pharmacological treatment (Table 4).

### 3.2. Sporothrix spp. Identification

Thirty-nine clinical strains were identified as *S. brasiliensis*, three as *S. globosa* and one as *S. schenckii*. These clinical strains showed 99–100% similarity with sequences deposited in the GenBank database (Figure 1). The obtained sequences for the Calmodulin gene during this study were deposited in GenBank under the accession numbers GU456632, and OQ673213–OQ673254.

### 3.3. Antifungal Susceptibility Profile

Considering that there are no defined clinical breakpoints for *Sporothrix* spp., the results of this study were evaluated according to the epidemiological cutoff values (ECVs) for *S. brasiliensis* and *S. schenckii*. All the 39 *S. brasiliensis* strains were classified as wild type (WT) for TRB and POS. However, one *S. brasiliensis* strain was classified as non-WT for AMB (MIC = 8 µg/mL) and another strain was classified as non-WT for ITR (MIC = 8 µg/mL). *Sporothrix schenckii* was classified as non-WT for all the antifungals tested (Table 5). For *S. globosa*, there are no defined ECVs.

For ISA, the highest MIC value found was 4 µg/mL for all the *Sporothrix* species as follows: *S. brasiliensis* (N = 19/39), *S. globosa* (N = 1/3) and *S. schenckii* (N = 1/1).

### 3.4. Description of Clinical Data of Patients According to the Type of Fungal Species Identified

Among the 21 cases reporting contact with plants and/or soil, *S. brasiliensis* (19 cases), *S. globosa* (one case) and *S. schenckii* (one case) predominated. In the 12 cases of sporotrichosis acquired from an unknown source, *S. brasiliensis* (10 cases) and *S. globosa* (two cases) were the identified species. Furthermore, *S. brasiliensis* was identified in six other distinct forms of transmission. In a specific case of sporotrichosis after inoculation with fishbone in the distal phalanx of the 2nd left finger of a 48-year-old female patient, the lymphocutaneous clinical form occurred, and *S. brasiliensis* was identified as the causative agent of the disease.

The *Sporothrix schenckii* 43461 strain was classified as non-WT for the antifungal drugs AMB, TRB, ITR and POS. This strain was isolated in 2012 from a 71-year-old female patient reporting contact with plants and/or soil. The patient stated that she had had no previous contact with animals. The patient manifested lymphocutaneous sporotrichosis. She required seven months of treatment with ITR 100 mg/day and progressed to clinical cure.

The *Sporothrix brasiliensis* 48016 strain was classified as non-WT for ITR. This strain was isolated in 2015 from a 22-year-old female patient, with a complete higher education, working as a tax assistant. The patient stated that she had had no previous contact with animals. She presented with the fixed cutaneous form. The treatment was carried out with ITR for four months, initially with a dosage of 100 mg/day and at the end of treatment with 200 mg/day, with clinical cure.

Finally, the *S. brasiliensis* 49753 strain was classified as non-WT for AMB. This strain was isolated in 2017 from a 41-year-old male patient, who worked as a physical education teacher, where he had a backyard and constantly manipulated plants and/or soil. In addition, this patient had no domestic animals and reported no contact with animals. He presented the lymphocutaneous form in the upper limb and required 1 month of treatment with ITR 200 mg/day, followed by 3 additional months with 100 mg/day, to obtain clinical cure.

Among the three strains of *S. globosa*, one was acquired after contact with plants and two were isolated from patients who did not know how the disease was acquired and stated that they did not own or have contact with any animal. The single *S. schenckii* strain identified in this study was isolated from a patient reporting hiking and trauma with plants and/or soil contact. Moreover, he stated that he did not own or have contact with cats (Table 6).

## 4. Discussion

Sporotrichosis has emerged as an important fungal disease due to changes in its epidemiology, geographic distribution and taxonomy [17].

In Rio de Janeiro, Brazil, there is a predominance of zoonotic sporotrichosis cases in female patients, probably due to the high number of women who are housewives or work at home and are in contact with sick cats [10]. However, in this study, focused on non-zoonotic transmission sporotrichosis, we observed a predominance of male patients after trauma with plants and soil contact, which is in accordance with the data related in the scientific literature [18]. Students, housewives and retirees were the most affected groups among the cases of non-zoonotic sporotrichosis, likely due to exposure to *Sporothrix* during leisure activities. These patients reported no contact with animals. Additionally, different occupations related to masonry, civil works and contact with plants, soil and decaying wood have been associated with non-zoonotic transmission sporotrichosis. This is the classic form of acquisition traditionally observed in specific occupational populations, such as agricultural workers and gardeners [10].

Among the pathogenic species causing human sporotrichosis, *S. schenckii* [8] and *S. globosa* [8,19] are usually associated with non-zoonotic transmission and *S. brasiliensis* with zoonotic transmission [20]. In this study, *S. brasiliensis* was the most isolated species followed by *S. globosa* and *S. schenckii*. 

Regarding the source of infection, most patients reported trauma involving plants and/or soil. Soil handling, gardening and contact/trauma with plants, such as wood branches and rose bushes, are classical sources described for acquiring the disease. Surprisingly, these cases were caused by *S. brasiliensis*, but there were also cases caused by *S. schenckii* and *S. globosa*.

The first environmental isolation of *S. brasiliensis* was reported in Rio de Janeiro state from a wood sample in an area with recurrent cases of human and feline sporotrichosis for more than 10 years [21]. Subsequently, *S. brasiliensis* DNA was identified in soil samples in a rural area in Rio de Janeiro, suggesting the survival of *S. brasiliensis* in the environment and reinforcing the likelihood of the occurrence of other transmission forms from environmental sources [22]. *Sporothrix brasiliensis* has also been isolated in feces collected from the intestines of necropsied cats and from a pile of sand, suggesting that the feces of sick cats can contaminate the soil, creating new environmental reservoirs [23]. Another potential source of soil contamination could be the improper disposal of carcasses from animals that died due to sporotrichosis [24]. The nineteen *S. brasiliensis* strains associated with plants and/or soil in this study reinforce the potential for acquiring sporotrichosis from environments contaminated with this species. 

In 2002, a case of sporotrichosis due to insect bites was reported [25]. Silva et al. (2012) reported insect bites, birds interactions or accidents with dogs, corresponding to 10 reports (0.5%) among the 1848 human sporotrichosis cases [10]. Although insect bites as a source of sporotrichosis infection are scarce, our study detected four (9.4%) cases due to *S. brasiliensis* related to insect bites, including three involving mosquitoes and one with a “wasp”. It is important to mention that cases of sporotrichosis caused by insect bites could be misdiagnosed as leishmaniasis, highlighting the role of laboratory diagnosis.

A case of sporotrichosis due to fishbone trauma was reported in this study. Haddad et al. described an 18-year-old fisherman injured by spines from the dorsal fin of a fish (*Cichlidae*) in his 3rd left finger, forming an ascending nodular lymphangitic lesion [26]. Possibly, the cause of infection through fish may be due to the presence of *Sporothrix* spp. in aquatic environments. The environmental detection of *S. brasiliensis* DNA in a stream and beside a river in Rio de Janeiro has been previously reported [22]. Furthermore, this study reports unusual forms of *Sporothrix* spp. infection following human aggression or bite, trauma with an iron gate and previous skin injury. Another non-zoonotic source of *S. brasiliensis* infection was described by Fichman et al. (2022), where two cases of sporotrichosis were reported after tattoos [11].

The role of natural sources of *S. brasiliensis* infection for humans and cats remains poorly understood. However, the recent hyperendemic condition may relate to the fungus’s complex adaptive evolutionary strategies [27]. The literature suggests that *S. brasiliensis* is an emerging species evolutionarily more adapted to mammalian parasitism compared to *S. schenckii* [28]. Nevertheless, *S. globosa* was more prevalent than *S. schenckii* in this study. Despite its lower thermotolerance and virulence, S. globosa may have additional virulence factors contributing to disease even in areas with more virulent species. 

In this study, the most common clinical form was lymphocutaneous. *Sporothrix brasiliensis* was associated with all clinical manifestations, including severe forms related to plant and/or soil infection. *Sporothrix globosa* was associated with the lymphocutaneous and fixed cutaneous forms, and *S. schenckii* with the lymphocutaneous form. The literature associates *S. brasiliensis* with lymphocutaneous and fixed cutaneous forms [2], as well as with severe cases and atypical manifestations [29], all via zoonotic transmission [2,30,31].

Bernardes-Engemann et al. (2022) evaluated 100 *S. brasiliensis* strains over 20 years during the hyperendemic scenario in Rio de Janeiro state, with 13 strains classified as non-WT [30,31]. In our study, only 3 of the 43 clinical strains were classified as non-WT, including 2 strains of *S. brasiliensis* and 1 of *S. schenckii*. All the patients infected with these strains received ITR (100 mg and 200 mg/day) and the duration of treatment was four and seven months for *S. brasiliensis* and *S. schenckii*, respectively. This finding is supported by the results obtained in the study conducted by Almeida-Paes, where *S. brasiliensis*-infected patients overall required shorter durations of ITR treatment compared to individuals with *S. schenckii* [29]. These results should be carefully analyzed, especially the in vitro results, since the host’s immune response plays an important role in the outcome of an infectious disease [21].

The lower MIC values for TRB and higher values for AMB were consistent with other studies [30,32]. Most *S. brasiliensis* isolates showed the same MIC values for ITR and POS (MIC = 1 µg/mL) and a higher value for ISA (MIC = 4 µg/mL). On the other hand, ISA inhibits the growth of *S. brasiliensis* resistant to ITR, which may be useful to treating sporotrichosis caused by resistant strains [33].

In conclusion, non-zoonotic sporotrichosis still occurs in Rio de Janeiro state, Brazil, with *S. brasiliensis* being the main etiological agent causing infection after trauma with plants or soil contact, followed by *S. globosa* and *S. schenckii*. Moreover, non-WT strains were found, which indicates the need for continuous surveillance of the antifungal susceptibility profile of these etiologic agents. It is important to keep investigating other natural sources of *S. brasiliensis* to better understand the agent and its environment and host cycle.

## Figures and Tables

**Figure 1 jof-10-00610-f001:**
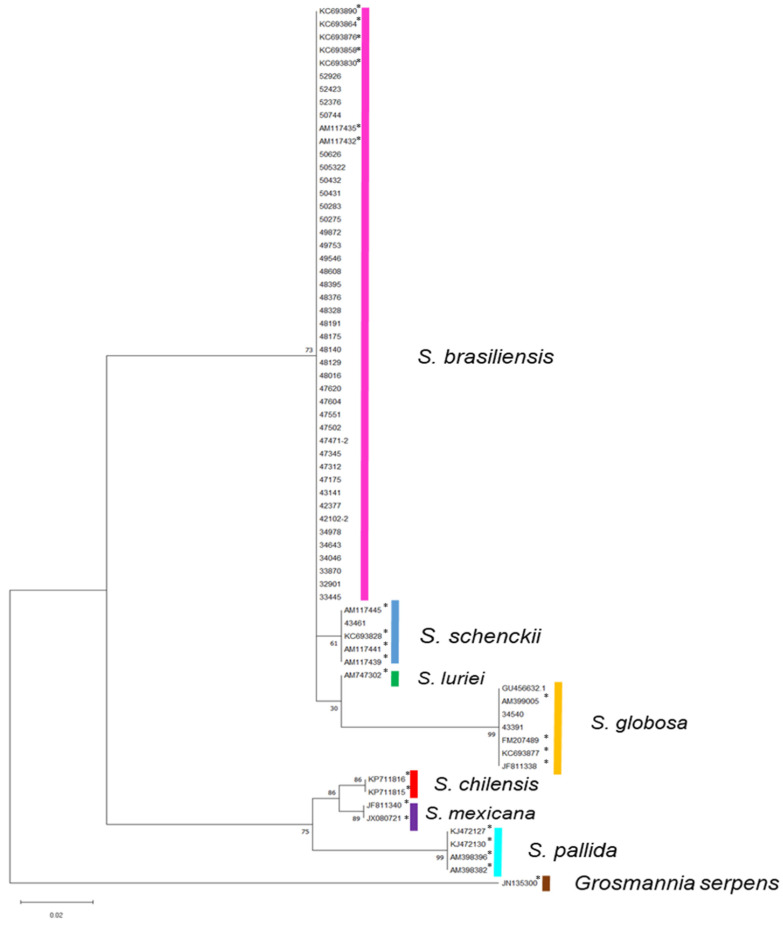
Phylogenetic tree based on partial sequencing of the gene encoding calmodulin with 43 clinical strains and 25 (*) sequences from GenBank. Maximum likelihood method. Significance verified using the bootstrap confidence test with 1000 repetitions indicated in the nodes formed in the phylogenetic tree.

**Table 1 jof-10-00610-t001:** Epidemiological, clinical, therapeutic and outcome data of 43 patients with non-zoonotic sporotrichosis followed at INI/Fiocruz, Rio de Janeiro, Brazil, between 2005 and 2020.

Variable	Description	N (%)
Epidemiology	Plants and/or soil manipulation	21 (48.8%)
	Unknown	12 (27.9%)
	Insect bites	4 (9.4%)
	Previous trauma *	2 (4.7%)
	Human aggression	1 (2.3%)
	Human bite	1 (2.3%)
	Trauma with iron gate	1 (2.3%)
	Trauma with fishbone	1 (2.3%)
Clinical form	Lymphocutaneous	28 (65%)
	Fixed cutaneous	10 (23.3%)
	Cutaneous disseminated	2 (4.7%)
	Unifocal extracutaneous **	2 (4.7%)
	Disseminated	1 (2.3%)
Treatment	Itraconazole	27 (62.9%)
	Terbinafine	3 (7%)
Change in treatment	Itraconazole to terbinafine Itraconazole to amphotericin B Itraconazole to terbinafine to amphotericin B Itraconazole to posaconazole	4 (9.3%)
		1 (2.3%)
		1 (2.3%)
		1 (2.3%)
Adjuvant treatment	Cryosurgery associated with itraconazole Cryosurgery associated with itraconazole and terbinafine Cryosurgery associated with terbinafine Cryosurgery Warm water compress	2 (4.7%)
		1 (2.3%)
		1 (2.3%)
		1 (2.3%)
		1 (2.3%)
Outcome	Clinical cure	35 (81.4%)
	Lost to follow-up	6 (13.9%)
	Still under treatment	2 (4.7%)

* The patients confirmed that they had had contact with animals, especially cats. ** Extracutaneous: ocular and pulmonary involvement (one case each).

**Table 2 jof-10-00610-t002:** Clinical forms and lesion topography of the 43 patients with non-zoonotic sporotrichosis followed at INI/Fiocruz, Rio de Janeiro, Brazil, between 2005 and 2020.

Clinical Form (Total) (%)	Lesion Topography (Number of Patients) (%)
Lymphocutaneous (28) (65%)	Upper limbs (17) (39.5%)
	Lower limbs (6) (13.9%)
	Trunk (4) (9.3%)
	Face (1) (2.3%)
Fixed cutaneous (10) (23.3%)	Upper limbs (5) (11.6%)
	Lower limbs (4) (9.3%)
	Trunk (1) (2.3%)
Cutaneous disseminated (2) (4.7%)	Skin (2) (4.7%)
Disseminated (1) (2.3%)	Skin/Eye/Upper airways/Multifocal bones (1) (2.3%)
Unifocal extracutaneous (2) (4.7%)	Eye (1) (2.3%)
	Lung (1) (2.3%)

**Table 3 jof-10-00610-t003:** Description of clinical forms and occupation according to the age groups of the 43 patients with non-zoonotic sporotrichosis followed at INI/Fiocruz, Rio de Janeiro, Brazil, between 2005 and 2020.

Age Group	Clinical Form (N)	Occupation
0–18	Lymphocutaneous (5)	Students
	Fixed cutaneous (1)	
19–59	Lymphocutaneous (18)	Housewives; service workers and trade sellers; mid-level technicians; painting workers; service workers in general; teachers; masonry structure workers; bus drivers; auto mechanics; structural assembly workers in civil works; members of the armed forces, police and fire service; commerce operators in stores and markets; economists; and people outside the labor market.
	Fixed cutaneous (4)	
	Disseminated (1)	
	Cutaneous disseminated (1)	
	Unifocal extracutaneous (ocular) (1)	
≥60	Lymphocutaneous (5)	Retired; housewives; service workers and trade sellers, auto mechanics and painting workers
	Fixed cutaneous (5)	
	Cutaneous disseminated (1)	
	Unifocal extracutaneous (pulmonary) (1)	

**Table 4 jof-10-00610-t004:** Treatment and clinical outcomes of the 43 patients with non-zoonotic sporotrichosis followed at INI/Fiocruz, Rio de Janeiro, Brazil, between 2005 and 2020.

Treatment (Total)	Clinical Outcome (N)
Itraconazole (27)	Cure (22)
	Lost to follow-up (4)
	Still under treatment (1)
Terbinafine (3)	Cure (3)
Itraconazole to terbinafine (4)	Cure (4)
Itraconazole to amphotericin B (1)	Lost to follow-up (1)
Itraconazole to terbinafine to amphotericin B (1)	Lost to follow-up (1)
Itraconazole to posaconazole (1)	Still under treatment (1)
Cryosurgery (1)	Cure (1)
Cryosurgery associated with itraconazole (2)	Cure (2)
Cryosurgery associated with itraconazole and terbinafine (1)	Cure (1)
Cryosurgery associated with terbinafine (1)	Cure (1)
Warm water compress (1)	Cure (1)

**Table 5 jof-10-00610-t005:** In vitro susceptibility profile of the clinical strains of *Sporothrix* spp. isolated from patients with non-zoonotic sporotrichosis included in this study.

		MIC (µg/mL)			ECV Interpretation Number of Isolates	
Species (N)	Antifungal	Range	MIC_50_/MIC_90_	GM	WT	non-WT
	AMB	0.5–8	4/4	2.31	38	1
*S. brasiliensis*	TRB	0.03–0.12	0.06/0.12	0.06	39	0
(39)	ITR	0.5–8	1/2	1.11	38	1
	POS	0.25–2	1/2	1.11	39	0
	ISA	0.5–4	2/4	2.43	-	-
*S. globosa*	AMB	1–4	1/4	2.52	-	-
(3)	TRB	0.06–0.12	0.06/0.12	0.08	-	-
	ITR	1–2	1/2	1.26	-	-
	POS	1–2	1/2	1.59	-	-
	ISA	2–4	2/4	2.52	-	-
*S. schenckii*	AMB	8	-	-	0	1
(1)	TRB	0.25	-	-	0	1
	ITR	4	-	-	0	1
	POS	4	-	-	0	1
	ISA	4	-	-	-	-

MIC, minimum inhibitory concentration; AMB, amphotericin B; TRB, terbinafine; ITR, itraconazole; POS, posaconazole; ISA, isavuconazole; GM, geometric mean; WT, wild type; non-WT, non-wild type; ECV, epidemiological cutoff value for *S. brasiliensis* and *S. schenckii*.

**Table 6 jof-10-00610-t006:** Epidemiological and clinical data from patients infected with *S. globosa* and *S. schenckii* strains.

Strain	Sex/ Age	Occupation	Transmission	Clinical Form	Comorbidities	Treatment/Outcome
*S. globosa* 27135	F/77	Housewife	Contact with plants	Lymphocutaneous	-	ITR/Cure
*S. globosa* 34540	M/62	Auto mechanic	Previous trauma *	Fixed cutaneous	High blood pressure	ITR/TRB Cure
*S. globosa* 43391	F/71	Teacher	Previous trauma *	Lymphocutaneous	High blood pressure and diabetes mellitus	ITR/Cure
*S. schenckii* 43461	F/71	Retired	Contact with plants	Lymphocutaneous	Hyperthyroidism	ITR/Cure

* The patients confirmed that they had had no contact with animals, especially cats. TRB, terbinafine; ITR, itraconazole. *Sporothrix globosa* strains were isolated in 2005, 2008 and 2012, respectively. *Sporothrix schenckii* was isolated in 2012.

## Data Availability

The original contributions presented in the study are included in the article, further inquiries can be directed to the corresponding author.

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
