# Peer review of "Non-Zoonotic Transmission of Sporotrichosis: A Translational Study of Forty-Three Cases in a Zoonotic Hyperendemic Area"

_jof, 2024, doi:10.3390/jof10090610_

Round 1
Reviewer 1 Report
I have reviewed the manuscript submitted by Nahal J, et al which evaluated the epidemiological, clinical and therapeutic aspects of sporotrichosis with non-zoonotic transmission.
In general, there are many syntax, and grammar errors throughout that need to be corrected to better improve the understanding of the manuscript. Do not start sentences with “Regarding” or “concerning”.
The title, has nothing to do with the manuscript. It needs to be re-structured.
Abstract: has many syntax errors and run on sentences.
Introduction: syntax and grammar corrections.
Methodology:
Delete “Regarding” or “Concerning” from sentence, it is not used there to initiate a sentence.
Results:
3.4. What is practicing trails? Explain.
Discussion:
This section is too long for the data presented. Shorten by at least 33%.
In addition, the authors discuss epidemiological studies, but no environmental isolates are described.
Line 262-264: Authors have not presented any evidence regarding that speculation.
Line 300-302: Not sure where the authors get this suggestion. Not from this paper.
Tables:
Table 1-4 need to be re-structured, they are difficult to read and understand.
See above.
Author Response
- Review 1:
I have reviewed the manuscript submitted by Nahal J, et al which evaluated the epidemiological, clinical and therapeutic aspects of sporotrichosis with non-zoonotic transmission.
In general, there are many syntax, and grammar errors throughout that need to be corrected to better improve the understanding of the manuscript. Do not start sentences with “Regarding” or “concerning”.
Answer: Thanks for this comment. We correct what was requested in the review.
The title, has nothing to do with the manuscript. It needs to be re-structured.
Answer: Thanks for this comment. The title was re-structured as per your suggestion.
Abstract and Introduction: has many syntax and grammar errors and run on sentences.
Answer: Thanks for this comment. Abstract and Introduction was reviewed.
Methodology: Delete “Regarding” or “Concerning” from sentence, it is not used there to initiate a sentence.
Answer: Thanks for this comment. We deleted these sentences.
Results: 3.4. What is practicing trails? Explain.
Answer: Thanks for this comment. This sentence was excluded because it’s not necessary.
Discussion: This section is too long for the data presented. Shorten by at least 33%. In addition, the authors discuss epidemiological studies, but no environmental isolates are described.
Answer: Thanks for this comment. This section has been shortened as requested.
Line 262-264: Authors have not presented any evidence regarding that speculation.
Answer: Thanks for this comment. This reference was wrong. The correct reference was included.
Line 300-302: Not sure where the authors get this suggestion. Not from this paper.
Answer: Thanks for this comment. This information was excluded.
Tables: Table 1-4 need to be re-structured, they are difficult to read and understand.I have reviewed the manuscript submitted by Nahal J, et al which evaluated the epidemiological, clinical and therapeutic aspects of sporotrichosis with non-zoonotic transmission.
Answer: Thanks for this comment. All tables were re-structured.
Reviewer 2 Report
1. Are there any differences in clinical findings and treatment results between non-zoonotic sporotrichosis and zoonotic sporotrichosis by S. brasiliensis ?
2. Would you mention the annual incidence of non-zoonotic sporotrichosis by S. brasiliensis?
Compared to zoonotic sporotrichosis, the incidence of non-zoonotic sporotrichosis is very low. In Asia, the incidence of Sporotrichosis due to S. globose is decreasing. Environments may not be proper for S. globose. If S. brasiliensis can overcome the difficult environments, the non-zoonotic route will be important.
Please add the isolated year of each strain in Table 6.
Author Response
Rebutal letter
- Review 2:
- Are there any differences in clinical findings and treatment results between non-zoonotic sporotrichosis and zoonotic sporotrichosis by S. brasiliensis ?
Answer: It was not possible to make this comparison in our study. The isolates are all non-zoonotic.
- Would you mention the annual incidence of non-zoonotic sporotrichosis by S. brasiliensis?
Compared to zoonotic sporotrichosis, the incidence of non-zoonotic sporotrichosis is very low. In Asia, the incidence of Sporotrichosis due to S. globosa is decreasing. Environments may not be proper for S. globose. If S. brasiliensis can overcome the difficult environments, the non-zoonotic route will be important.
Answer: It was not possible to make this comparison because this study was retrospective.
Detail comments
Please add the isolated year of each strain in Table 6.
Answer: Thanks for this comment. We added this information in Table 6.
Reviewer 3 Report
In this work, Nahal et al. evaluated the clinical, epidemiological, and therapeutic aspects of 43 cases of sporotrichosis in humans in a hyperendemic region, whose transmission was not zoonotic. In addition, they evaluated the in vitro antifungal susceptibility profile for the strains isolated in the cases studied. The authors highlight that most non-zoonotic cases of sporotrichosis were caused by S. brasiliensis, which is the main etiological agent in the area and traditionally associated with zoonotic transmision, followed by S. globosa and, finally, only one case by S. schenckii, which has been the etiological agent most frequently associated with non-zoonotic infections. Likewise, based on the antifungal susceptibility profile, two strains of S. brasiliensis and one strain of S. schenckii were classified as non-WT strains due to the resistance they presented to the antifungals studied. The above is relevant because zoonosis is the main form of transmission of sporotrichosis in the study area, and most of the studies on the subject focus on this type of contagion. However, studying cases transmitted by the canonical route can also provide relevant information on the ecodistribution of the etiological agents, their routes of contagion, and the general form of the clinical manifestation.
The manuscript is well-structured and written clearly and concisely. The introduction provides sufficient information to contextualize the work's purpose and its relevance to the field. The methods are clearly described, although those that have already been described in other works and that are mentioned in this manuscript could be briefly summarized, mainly those that the authors mention having performed with modifications. The results are clearly described and presented in tables and figures with sufficient clarity. Likewise, the results are correctly discussed and allow the conclusions formulated by the authors to be supported. Although it may seem that the report of 43 cases is a small N from which to conclude, the study covers non-zoonotic cases that occurred over 15 years in a reference center, so this number of cases also allows us to infer the frequency with which these cases occur in a hyperendemic area. In general, I find it an interesting study that reflects important aspects of a mycosis that, in the study region, is usually analyzed mainly when the form of transmission is through contact with infected animals.
In section 2.3, Line 97, please briefly explain such modifications.
In Figure 1, I suggest using a different color or a mark to distinguish the control strains from the clinical isolates.
Author Response
Rebutal letter
- Review 3:
In section 2.3, Line 97, please briefly explain such modifications.
Answer: Thanks for this comment. DNA extraction was carried out with strains grown at 37ºC, in their yeast form, and it was not necessary to make changes to the protocol. We correct this in the text.
In Figure 1, I suggest using a different color or a mark to distinguish the control strains from the clinical isolates.
Answer: Thanks for this comment. We added “ * “ to differentiate reference strains.
Round 2
Reviewer 1 Report
Much improved and addresses the prior major stipulations and concerns from the initial submission. Still requires some minor grammatical and syntax improvements.
as above